# COVID-19 knowledge and mental health impact assessment in Haiti

Taehwan Park[1], Jean Pierre-Louis[2], Tachel Jean[3], Prachurjya Barua[3], Taheera T. Ilma[3], Mariana A. Pinanez[3], Joseph Ravenell[4], Chimene Castor[5] and Yolene Gousse[1] (iD)

[1]Department of Pharmacy Administration and Public Health, St. John's University, Queens, NY, USA; [2]Capracare, Inc., Les Cayes, Haiti; [3]Biomedical Program, College of Pharmacy and Health Sciences, St. John's University, Queens, NY, USA; [4]Department of Population Health, New York University Grossman School of Medicine, New York, NY, USA and [5]Department of Nutritional Sciences, Howard University, Washington, DC, USA

## Research Article

**Keywords:**
COVID-19; mental health; anxiety; stress; Haiti

**Corresponding author:**
Yolene Gousse;
Email: Goussey@stjohns.edu

## Abstract

Mental health is a significant public health challenge globally, and one anticipated to increase following the COVID-19 pandemic. In many rural regions of developing nations, little is known about the prevalence of mental health conditions and factors that may help mitigate poor outcomes. This study assessed the impact of the COVID-19 pandemic on mental health and social support for residents of rural Haiti. Data were collected from March to May 2020. The Patient Health Questionnaire subscales for anxiety and depression, and the Perceived Stress Scale were utilized in addition to tailored questions specific to COVID-19 knowledge. Half (51.8%) of the 500 survey respondents reported COVID-19-related anxiety and worrying either daily or across a few days. Half (50.2%) also reported experiencing depression daily or across several days. Most (70.4%) did not have any social support, and 28.0% experienced some stress, with 13.4% indicating high perceived stress. Furthermore, 4.6% had suitable plumbing systems in their homes. The results were immediately actionable, informing the implementation of a mental health counseling program for youth following a loss of social support through school closures. Long-term investments must be made as part of public health responses in rural communities in developing nations, which remain under-studied.

## Impact statement

This study presents a first-of-its-kind assessment of the prevalence of mental health outcomes, social support and economic hardship for residents of rural Haiti after the onset of the COVID-19 pandemic. Mental illness is a significant global health issue, and less is known about its challenges in low- and middle-income nations. In Haiti, a confluence of factors, including persistent poverty and limited access to health care, have been exacerbated by the COVID-19 pandemic. Emerging data suggest the pandemic has increased mental health burdens, with the poorest communities being most affected. The study found that half of the 500 survey respondents (51.8%) reported symptoms of COVID-19-related anxiety and depression. Most respondents (70.4%) reported having no social support. Of those reporting some level of stress (28%), 13.4% reported high perceived stress. This demonstrates that the COVID-19 pandemic has had significant impacts on residents of rural communities in lower-income countries, which have not been included in much of the current discourse on the pandemic. Place of stay and facility for washing hands were found to be significantly associated with anxiety, depression and stress, Wilks' Lambda ≤0.05. The latter has implications for social deprivation factors being the most significant drivers of mental status. Study results were immediately actionable by the participating community-based organization (CBO), which implemented a youth counseling program providing social support and explored additional opportunities to address mental health needs. The study's outcomes and model can stimulate interest among researchers and CBOs to forge partnerships and design tailored health education initiatives for similar underserved communities worldwide. Public health practitioners must identify unique mental health needs of residents in diverse communities and use insights to develop tailored outreach and intervention programs. It is important that the needs of residents in these low- and middle-income nations do not go unfunded and unresearched during global pandemics.





## Introduction

Mental illness is a significant global health burden which, combined with addictive disorders, affects more than a billion people worldwide (Rehm and Shield, 2019). Individuals experiencing mental illness during their lifetime have a shorter life expectancy by an average of 10.1 years, compared to those without mental illness (Walker et al., 2015; Deeg et al., 2018). Worldwide, mental health disproportionately affects certain populations based on factors, including access to services, stigma of seeking care and poverty. In low- and middle-income nations, extreme poverty, lack of educational and economic opportunities, hunger and overcrowded or unsafe living conditions together aggravate and lead to increased rates of mental illness (Lund et al., 2011). Mental illness and poverty often exacerbate a negative cycle in which persons who experience social deprivation are at higher risk for developing mental illness, and those with mental illness are at higher risk for poor daily functioning and lower quality of life (Kumar and Kumar, 2020).

Haiti is a low- to middle-income nation in the Caribbean with over 11 million residents. Many Haitians experience poor health outcomes, including mental health, due to Haiti's under-resourced health care system, struggling economy and ongoing political unrest (Quran, 2019). The 2010 earthquake in Haiti caused high morbidity and mortality rates, with nearly 57% of deaths attributed to communicable diseases such as tuberculosis, respiratory conditions and diarrheal-related diseases. The aftermath of the earthquake also further exacerbated existing mental health issues in Haiti, including increasing the rate of depression and anxiety (Wagenaar et al., 2012), and of the population experiencing post-traumatic stress disorder (PTSD), and 50% experienced a major depressive disorder (Castle, 2020).

Approximately 6.4 million (59%) Haitians live below the poverty line, with nearly a quarter (24%) of the population living in extreme poverty (IFRC, 2019). Furthermore, only 36% of the population has access to electricity, 56% has access to clean water and only 28% has access to basic sanitation (Louis-Jean et al., 2020). These social determinants of health contribute to social deprivation, which represents limited access to society's resources due to poverty, discrimination, or other disadvantages that are associated with poor mental health outcomes.

In Haiti, as of January 3, 2023, 34.4% of the confirmed COVID-19 cases resulted in death (WHO, 2023), compared to the 1.5% of the confirmed COVID-19 cases dying in early March 2020 (OCHA, 2020), indicating a rapid increase in the COVID-19 mortality rate. Emerging data have shown that the pandemic has increased mental health burden (Panchal et al., 2020; Lozano et al., 2021). In low-income countries, the poorest communities were most affected by COVID-19 (Prates and Barbosa, 2020) and had a disproportionately high suicide rate (Pirkis et al., 2021). To date, there is little research examining the effects of COVID-19 on mental health in rural communities of low- and middle-income nations, including in rural Haiti. Identifying the mental health and social impacts of COVID-19 on Haitians will be important to accurately inform public health programs and policies designed to address heightened disparities.

In the current study, the authors assessed the impact of the COVID-19 pandemic on mental health, social support and economic hardship and identified areas of need for rural community members in Haiti after the onset of the COVID-19 pandemic. Through a partnership with a CBO, a mental health assessment was implemented for the first time in this community, to the authors' knowledge. The authors also share strategies to address behavioral health and related social needs based on the study's supported findings.

## Methods

The present study used a cross-sectional survey design developed through a collaborative partnership between capra*care Inc.*, a CBO in Haiti and St. John's University (SJU) Public Health Program in Queens, New York, in the United States (US). This partnership builds on a 5-year history of research partnership, which has included prior research studies and capra*care* serving as a fieldwork host site for SJU public health graduate students to develop and implement programs.

The organization capra*care*, founded in 2009, located in the rural community of Fonfrede, Haiti, provides basic medical and preventive health care for women and children, health education, social services, professional development training and low-threshold mental health counseling. All services delivered by capra*care* are free or offered at a low cost for community residents. Fonfrede, Haiti, a community located 120 miles outside the capital, Port-au-Prince, has an estimated population of 20,000 residents. In 2009, more than 40% of the Haitian population reported not accessing the health system due to their inability to afford the associated costs (Maselko, 2017). As a result of lack of access to health care, community members have learned to rely on alternative medicine and religious healers to address illnesses and manage diseases (Materu et al., 2020).

SJU, founded in 1870, emphasizes strong ties to local communities in Queens and surrounding regions and facilitates national and international initiatives. The SJU faculty member who served as the Principal Investigator for the project was not previously affiliated with capra*care*'s health care or mental health programs.

All project materials were approved by St. John's University Review Board (FWA # 00009066).

All data used in this study was collected between March and May 2020.

### Recruitment and data collection procedures

Respondents were recruited by capra*care* staff and Community Health Workers (CHWs) who canvased Fonfrede and Les Cayes in Haiti to identify potential study participants. The door-to-door canvassers knocked on household doors and explained the purpose of the study, identified potentially eligible participants and ensured a representative sample. The eligible study population comprised of persons who identified as (1) adults aged 18 years or older; (2) residents of Haiti at the time of data collection; and (3) those who understood and spoke Haitian Creole. Staff of capra*care* and CHWs explained the project to potential study participants. Eligible individuals interested in this study were provided with a set of assessment questions before survey deployment. All respondents provided verbal consent prior to being formally enrolled in the assessment.

Face-to-face interviewer-led interviews with enrolled community members were conducted. Participant responses were recorded by interviewers, with periodic checks of interviews conducted to check for completion, accuracy and integrity of survey completion and for interviewer bias. The data were collected and centrally organized and stored to ensure confidentiality and protection prior to analysis.

## Survey instruments

The survey instruments used in this study were developed in English by project staff and translated to Haitian Creole by capra-*care's* team. To establish equivalence, clarity and cultural appropriateness of the survey, the English version was translated to Haitian Creole and then back-translated into English by professional translators proficient in both languages. The survey was pilot tested with a sample of individuals representative of the priority population. The survey comprised both open and closed-ended questions, collecting data on five construct areas: (1) Respondent sociodemographic factors; (2) COVID-19 knowledge and ability to care for persons diagnosed with the virus; (3) Domestic composition and structural factors; (4) Existing social network as a support system and resource; and (5) COVID-19 impact on mental health status.

## Sociodemographic factors

The following standard sociodemographic variables (Table 1) were assessed to describe the study sample: age, gender, nativity, region of residence in Haiti, marital status, employment status, educational status and household composition. One additional question

**Table 1.** Study respondent characteristics (*n* = 500)

| Characteristics | N | % |
|---|---|---|
| Gender | | |
| Male | 169 | 33.80 |
| Female | 307 | 61.40 |
| Not reported | 24 | 4.80 |
| Nativity | | |
| Haiti | 495 | 99.00 |
| Not born in Haiti | 3 | 0.60 |
| Not reported | 2 | 0.40 |
| Age | | |
| 18–85 | 469 | 93.80 |
| Not reported | 31 | 6.20 |
| Region of residence in Haiti | | |
| Kay or Les Cayes | 448 | 89.60 |
| Tobek | 4 | 0.80 |
| Fontfrede (K-Sowoz) | 4 | 0.80 |
| Not reported/unknown | 44 | 8.80 |
| Marital status | | |
| Married | 115 | 23.00 |
| Domestic partnership | 8 | 1.60 |
| Divorced | 1 | 0.20 |
| Single | 358 | 71.60 |
| Not reported/unknown | 18 | 3.60 |
| Employment status | | |
| Employed | 76 | 15.20 |
| Unemployed | 392 | 78.40 |
| Not reported/unknown | 32 | 6.40 |

(*Continued*)

**Table 1.** (*Continued*)

| Characteristics | N | % |
|---|---|---|
| Education | | |
| Did not complete high school | 341 | 68.20 |
| High school graduate/GED or Equivalent | 44 | 8.80 |
| Completed some college/2 yr college or vocational/technical/trade school | 83 | 16.60 |
| College graduate (4 years or higher) | 22 | 4.40 |
| Not reported/unknown | 10 | 2.00 |
| Household composition | | |
| Alone | 54 | 10.80 |
| Living with someone | 443 | 86.60 |
| Not reported/unknown | 3 | 2.60 |
| Received services from capra*care*? | | |
| Yes | 299 | 59.80 |
| No | 201 | 40.20 |
| Have you heard of COVID-19 before today? | | |
| Yes | 485 | 97.00 |
| No | 3 | 0.60 |
| Not reported/unknown | 12 | 2.40 |
| Where did you first hear about COVID-19? | | |
| Family/friend | 38 | 7.60 |
| Social media | 85 | 17.00 |
| Radio/TV | 334 | 66.80 |
| capra*care* | 26 | 5.20 |
| Church | 16 | 3.20 |
| Not reported/unknown | 1 | 0.20 |
| Can you isolate yourself or family member sick with COVID-19? | | |
| Yes | 302 | 60.40 |
| No | 178 | 35.60 |
| Not reported/unknown | 20 | 4.00 |
| Do you know how to care for yourself or family member sick with COVID-19? | | |
| Yes | 301 | 60.20 |
| No | 182 | 36.40 |
| Not reported/unknown | 17 | 3.40 |

asked whether the respondent had received services from capra*care* (yes/no).

## COVID-19 knowledge and isolation strategies

Four questions assessed respondent's knowledge about the COVID-19 pandemic and isolation strategies. The questions included: *(1) Have you heard of COVID-19 before today? (2) If yes, where did you first hear about COVID-19? (3) Can you isolate yourself or a family member sick with COVID-19 in your household? All response options were dichotomous (yes/no).*

**Table 2.** Domestic and social network support (*n* = 500)

| Where do you live or stay most of the time? | | |
|---|---|---|
| Response | N | % |
| In my own home | 406 | 81.20 |
| Parent's or family member's house | 39 | 7.80 |
| At work | 1 | 0.20 |
| Homeless/on street | 51 | 10.20 |
| Not reported/unknown | 3 | 0.60 |
| Facilities you have for washing hands in your household | | |
| Well | 122 | 24.40 |
| Makeshift container such as a bucket | 249 | 49.80 |
| Plumbing system | 23 | 4.60 |
| Fetch water/public tap | 104 | 20.80 |
| Not reported/unknown | 2 | 0.40 |
| What type of latrine is available in your household? | | |
| Outdoor/outhouse | 440 | 88.00 |
| Indoor toilet | 46 | 9.20 |
| Other – on the floor | 14 | 2.80 |
| Can someone in your social network provide care if you contracted COVID-19? | | |
| **Response** | **N** | **%** |
| Yes | 149 | 28.40 |
| No | 345 | 70.40 |
| Not reported/unknown | 6 | 1.20 |
| For yes responses, social network relationship (*n* = 149) | | |
| Spouse | 28 | 18.70 |
| Parent/s | 50 | 33.60 |
| Other (children, sibling, friend/s) | 71 | 47.70 |

### Domestic conditions and social support network

Multiple variables validated in prior research were included in the survey to measure household living conditions (Dias and de Oliveira, 2018; Table 2). Respondents were asked about their living facilities, facilities for washing hands in their household and type of latrine available in their household. One question with a dichotomous response (yes/no) asked whether respondents had someone who could provide care if they contracted COVID-19. Those who responded 'yes' were further asked to specify who could provide the care. The responses to this follow-up open-ended question were categorized based on the relationship to the participant for evaluation purposes (e.g., spouse, parent(s)).

### Mental health status

To assess mental health status, multiple standardized questionnaires were used, including the Patient Health Questionnaire (PHQ-4) and the Perceived Stress Scale (PSS). Both scales have been broadly used in both clinical and nonclinical settings due to their strong reliability and validity (Kroenke et al., 2009; Materu

et al., 2020). In this study, two PHQ-4 subscales were used (anxiety and depression) and summed to create total scores. The PHQ-4 questions used in this study were adapted to be COVID-19 period-specific and were modified with the following language to read: 'Over the last two weeks (*during COVID-19*), how often have you been bothered by the following problems?' (1) Feeling nervous, anxious or on edge *due to COVID-19;* (2) Not being able to stop or control worrying *due to COVID-19*; (3) Feeling down, depressed or hopeless *due to COVID-19*; and (4) Little interest or pleasure in doing things *due to COVID-19*. Likert responses ranged from not at all to more than half the days. The PHQ-4 total score ranges from 0 to 12, with categories of psychological distress being: none (0–2), mild (3–5), moderate (6–8) and severe (9–12). A total score of ≥3 for the first two questions suggests anxiety. A total score of ≥3 for the last two questions suggests depression (Kroenke et al., 2009) The PHQ-4 scale is a validated scale with a Cronbach Alpha of $\alpha = 0.81$ which indicates good internal consistency (Materu et al., 2020). No changes to the scoring or scales of the PHQ-4 were made.

The PSS-4 measures the degree to which situations in one's life over the past month are appraised as stressful. The instrument's questions were adapted to be COVID-19 period-specific by adding the following language to questions. The PSS-4 questions read 'In the last month, (1) how often have you felt that you were unable to control the important things in your life *due to COVID-19*; (2) how often have you felt confident about your ability to handle your personal problems *due to COVID-19*; (3) how often have you felt that things were going your way *due to COVID-19*; and (4) how often have you felt difficulties were piling up so high that you could not overcome them *due to COVID-19*'. Likert responses for each item ranged from never to very often with scores from 0 to 4. Higher scores reflect higher degree and longer duration of perceived stress (Vallejo et al., 2018). Although the PSS-4 has a Cronbach alpha of <0.7 indicating low internal validity, it is useful and feasible in situations where a short questionnaire is required (Lee, 2012). The responses to the PSS-4 questions were summed to generate total PSS scores, which were further categorized into low, moderate and high perceived stress. Neither the PSS-4's scale nor scoring were changed.

To supplement the PSS-4, four questions were developed to capture the levels of and changes in stress experienced during the pandemic. One question was intended to compare perceived stress during the pandemic to that before the outbreak. Answer choices for this question were categorical, including 'More stress than usual; same level of stress as usual; or less stress than usual'. Respondents who indicated 'more stress than before' were asked for more explanation. Four questions (yes/no response) examined the type of stress, including 'worry about overall financial state'; 'the ability to afford food'; 'worry about becoming sick with COVID-19'; and 'worry about family member becoming sick with COVID-19'.

### Statistical analyses

All variables included in the study were examined using descriptive statistics. Bivariate Chi-square tests were used to measure associations of perceived stress, anxiety and depression with the explanatory variables in the study. The study respondents were first described in terms of socio-demographics using univariate statistics. The sample was then further described in terms of COVID-19 experiences, social support, household conditions and COVID-19 knowledge. Mental health scores were derived

for each element of stress and examined using Chi-square tests in association with all explanatory factors. In addition, Generalized Linear Model (GLM) Multivariable analysis was performed to analyze the relationships between multiple independent factors to assess their effect on mental health status (anxiety, depression and perceived stress). All statistical analyses were performed using IBM SPSS at the $\alpha < .05$ level.

## Results

The study sample comprised 500 adults aged 18–85 years with a mean age of 39 (sd = 17). Table 1 describes the respondents' sociodemographic characteristics, prior treatment at capra*care* and current knowledge of COVID-19. Nearly all study respondents (99%) were born in Haiti; 61.4% were females. The majority lived in Kay or Les Cayes in Haiti (89.6%), were single (71.6%) and unemployed (78.4%). Approximately 70% of the respondents did not complete high school and 17% completed a 2-year college or vocational/technical training school. Most (86.6%) reported living with someone such as a spouse, parents, grandparents, aunt and/or children. Over half of the study respondents (59.8%) had received services at capra*care*.

Almost all respondents (97%) reported having prior knowledge of COVID-19. The majority reported learning about the pandemic via radio/television (66.8%) or through social media (17%). Approximately 60% reported that they had the means to isolate themselves if they contracted COVID-19 and had knowledge of how to care for someone sick with COVID-19.

### Domestic and social network support

The domestic characteristics and social network support available to the respondents are presented in Table 2. While the majority (81.2%) indicated they lived in their own home, 10.2% were homeless. When asked about the availability of an in-home washing facility, 49.8% reported using a makeshift container or bucket to store water and wash hands, 24.4% reported using unfiltered water from a well and 20.80% reported relying on public taps or a public community well. Only 4.6% reported that having a suitable plumbing system in their home, while the large majority (88%) indicated using an outdoor latrine facility or an outhouse.

Over a quarter of respondents (28.4%) reported having a supporting social system in case of a COVID-19 infection. Most (70.4%) did not have any familial or social support available. The support system comprised of parents (33.6%), spouses (18.7%), and children, siblings and/or friends (47.7%).

### Mental health during the COVID-19 pandemic

PHQ-4 questions and the respondents' scores on mental health status are shown in Table 3. When asked about feelings of helplessness or anxiety due to the pandemic, nearly one-third reported experiencing anxiety for several days and 20% reported feeling it nearly every day. In addition, while 43% of the respondents reported 'no worry', 30.4% were worried for a few days and approximately 21.4% were worried almost every day. When asked about feeling depressed or hopeless, nearly one-third (29%) were depressed for several days and 21.2% were depressed

**Table 3.** Mental health status – PHQ-4 and perceived stress during COVID-19 (n = 500)

| Mental health status – PHQ-4 | | |
|---|---|---|
| Response | N | % |
| How often have you felt nervous, anxious or on edge due to COVID-19? | | |
| Not at all | 213 | 42.60 |
| Several days | 148 | 29.60 |
| More than half the days | 22 | 4.40 |
| Nearly every day | 111 | 22.20 |
| Not reported/unknown | 6 | 1.20 |
| How often do you worry or are unable to stop worrying due to COVID-19? | | |
| Not at all | 215 | 43.00 |
| Several days | 152 | 30.40 |
| More than half the days | 22 | 4.40 |
| Nearly every day | 107 | 21.40 |
| Not reported/unknown | 4 | 0.80 |
| How often do you feel down, depressed or hopeless due to COVID-19? | | |
| Not at all | 211 | 42.20 |
| Several days | 145 | 29.00 |
| More than half the days | 33 | 6.60 |
| Nearly every day | 106 | 21.20 |
| Not reported/unknown | 5 | 1.00 |
| How often do you have little interest or pleasure in doing things? | | |
| Not at all | 224 | 44.80 |
| Several days | 145 | 29.00 |
| More than half the days | 36 | 7.20 |
| Nearly every day | 90 | 18.00 |
| Not reported/unknown | 5 | 1.00 |
| PHQ4 Depression Subscale Score | | |
| Depression | 169 | 33.80 |
| No depression | 321 | 64.20 |
| Not reported/unknown | 10 | 2.00 |
| PHQ4 Anxiety Subscale Score | | |
| No anxiety (score less than 3) | 342 | 68.40 |
| Anxiety (score of 3 or higher) | 153 | 30.60 |
| Not reported/unknown | 5 | 1.00 |
| Perceived Stress Scale (PSS-4) | | |
| In the last month, how often have you felt that you were unable to control the important things in your life due to COVID-19? | | |
| Never | 183 | 36.60 |
| Almost never | 122 | 24.40 |
| Sometimes | 53 | 10.60 |
| Fairly often | 140 | 28.00 |
| Very often | 0 | 0.00 |
| Not reported/unknown | 2 | 0.40 |

(Continued)

**Table 3.** (*Continued*)

| Mental health status – PHQ-4 | | |
|---|---|---|
| Response | *N* | % |
| In the last month, how often have you felt confident about your ability to handle your personal problems due to COVID-19? | | |
| Never | 195 | 39.00 |
| Almost never | 118 | 23.60 |
| Sometimes | 46 | 9.20 |
| Fairly often | 137 | 27.40 |
| Very often | 1 | 0.20 |
| Not reported/unknown | 3 | 0.60 |
| In the last month, how often have you felt that things were going your way due to COVID-19? | | |
| Never | 158 | 31.60 |
| Almost never | 21 | 4.20 |
| Sometimes | 212 | 42.40 |
| Fairly often | 14 | 2.80 |
| Very often | 88 | 17.6 |
| Not reported/unknown | 7 | 1.40 |
| In the last month, how often have you felt difficulties were piling up so high that you could not overcome them due to COVID-19? | | |
| Never | 44 | 8.80 |
| Almost never | 15 | 3.00 |
| Sometimes | 199 | 39.80 |
| Fairly often | 20 | 4.00 |
| Very often | 221 | 44.20 |
| Not reported/unknown | 1 | 0.20 |
| PSS Score – Stress level scores (*n* = 359) | | |
| Low perceived stress (1–4) | 29 | 8.00 |
| Moderate perceived stress (5–9) | 282 | 78.60 |
| High perceived stress (10–13) | 48 | 13.40 |
| Complementary Perceived Stress Questions | | |
| Compared to before the COVID-19 outbreak, how stressed do you feel now? | | |
| More stress than usual (specify) | 154 | 30.80 |
| Same level of stress as usual | 155 | 31.00 |
| No stress | 174 | 34.80 |
| Not reported/unknown | 17 | 2.40 |
| What is your primary source of stress? (*n* = 309) | | |
| COVID-19 | 68 | 22.00 |
| Lack of security/situation in the country | 4 | 1.29 |
| Thought of going to the hospital/sickness | 2 | 0.67 |
| Lack of healthcare | 1 | 0.32 |
| The news of sickness and death | 6 | 1.94 |
| I cannot do my activities | 2 | 0.66 |
| Not reported/unknown | 226 | 73.12 |

**Table 3.** (*Continued*)

| Mental health status – PHQ-4 | | |
|---|---|---|
| Response | *N* | % |
| Are you worried about your overall financial ability during the COVID-19 outbreak? | | |
| Yes | 402 | 80.40 |
| No | 96 | 19.20 |
| Not reported/unknown | 2 | 0.40 |
| Are you worried about your ability to afford food during the COVID-19? | | |
| Yes | 390 | 78.00 |
| No | 107 | 21.40 |
| Not reported/unknown | 3 | 0.60 |

nearly every day. Furthermore, about 30% experienced a loss of pleasure for several days and 18% reported losing pleasure nearly every day.

### Depression and anxiety scores

The majority (64.2%) were not depressed, but over one-third (33.8%) scored ≥3 on the PHQ-4 Depression Subscale, suggesting some level of depression (Kroenke et al., 2009; Materu et al., 2020). Similarly, most respondents (68.4%) scored ≤3 on the anxiety subscale, while nearly one-third (30.6%) scored ≥3 on the PHQ-4 anxiety, suggesting some level of anxiety.

### Perceived stress

The PSS-4 questions distribution, and analysis scores are reported in Table 3, along with the four complementary stress questions to gauge stress difference prior to and during the pandemic.

When asked how often they felt unable to control the important things in their life, most (61%) reported never or almost never, while only 28% reported fairly often. Survey respondents were also asked how often they felt confident in handling their personal problems due to COVID-19. Approximately 63% reported that they were always or almost always confident, and only 27% reported feeling confident rarely. When asked how often they felt that things were going their way, 42.4% responded 'sometimes', 17.6% reported 'very often', and 31.6% responded 'never'. Similarly, when asked how often they felt that difficulties were piling up beyond their ability, 44.2% reported 'very often', while 39.8% reported 'sometimes'.

PSS-4 scores for the sample ranged from 1 to 13. Approximately a quarter (28%) of the respondents experienced some level of stress. The majority (78.6%) had scores between 5 and 9, indicating moderate perceived stress; 13.4% had scores ranging 10–13, indicating high perceived stress. The remaining 8% scored between 1 and 4, indicating low perceived stress.

Comparing prepandemic stress levels to during the pandemic, 34.8% reported less stress, 31% reported the same stress level and 30.8% reported more stress. When asked to elaborate upon their primary source of stress, 22% attributed it to COVID-19, (thought or news of being sick and going to the hospital, lack of healthcare, inability to do personal activities). Another 3.59% indicated

**Table 4.** Bivariate associations between characteristics and mental health status (*n* = 500)

|  | Value | df | *P*-value |
|---|---|---|---|
| Worry that family become sick with COVID-19 and anxiety | | | |
| Pearson chi-square | 9.19 | 2 | 0.010 |
| Having family member who could provide care if someone became sick with COVID-19 and depression | | | |
| Pearson chi-square | 10.23 | 2 | 0.006 |
| Having someone who could provide care if a family became sick with COVID-19 infection and depression | | | |
| Pearson chi-square | 8.04 | 2 | 0.018 |
| Education level and perceived stress | | | |
| Pearson chi-square | 102.63 | 72 | 0.010 |
| Gender and anxiety | | | |
| Pearson chi-square | 6.245 | 2 | 0.044 |

suffering from stress unrelated to the pandemic (political unrest) with 1.29% indicating stress due to the lack of security and political climate. Questions related to finance and ability to purchase food showed that the majority (80.4%) were worried about their finances and their ability to purchase food (78%).

### Mental health associations

Bivariate associations between characteristics and mental health status were found to be significant (Table 4). A higher proportion of respondents who reported being worried about a family member contracting COVID-19 had anxiety scores of 3 or higher (34.7% vs. 21.3%, $\chi^2$ = 9.19, *p* = .01). Respondents who had a family member who could provide care had lower depression rates (20.7% vs. 37.9%) and lower anxiety rates than those without (22% vs. 34.2%, $\chi^2$=8.04, *p* = .02). PSS scores were also associated with education levels ($\chi^2$ = 102.63, *p* = .01). Furthermore, female respondents reported significantly higher anxiety levels than males (68% vs. 26.1%, reporting a score of 3 or higher) ($\chi^2$ = 6.25, *p* = .04).

### Multivariable analysis

Across all the tested covariates we find place of stay (Where do you live or stay most of the time?) and facility for washing hands (What kind of facilities do you have for washing hands in your household?) to be significantly associated with anxiety, depression and stress, each with Wilks' Lambda ≤0.05.

The facility for washing hands is associated with mild levels of stress (score ~ 5), anxiety (score 3.96 for Makeshift container) and depression (score 3.10 for Makeshift container) and with highest scores for 'Makeshift container for washing hands'. Similarly, places of stay, specifically those living outside or at work reported high levels of depression (score ≥ 3.0), anxiety (score ≥ 3.0) and mild stress (scores ~5).

### Discussion

The present study is one of the first to examine COVID-19 knowledge and the impacts of the pandemic on mental health among community members in rural areas of Haiti. Approximately half of the respondents had received services from capra*care* Inc., a CBO providing health and social service care to residents of the region. The prevalence of mental health outcomes identified among community members indicates the need for expanded services through community-based organizations to adequately reach remote residents and promote resiliency after the onset of the COVID-19 pandemic.

The study adds new data on mental health and social support prevalence in rural Haiti. Approximately one-fifth of the respondents reported not being able to isolate in their own home, increasing the risk of spreading COVID-19 and other communicable diseases. There is a need for education about preventive measures for COVID-19 and treatment plans in case of infection, tailored to residents of low- and middle-income nations who may not have access to isolation practices found in modernized communities.

Our study found high levels of COVID-19 awareness, indicating that news and health education effectively reach rural populations. This has positive implications for health education and awareness campaigns for COVID-19 and mental health efforts. The study found significant associations between having social support and reductions in both depression and anxiety, with lower rates of both conditions in persons with an identified source of support. Overall, the study respondents indicated moderate to high prevalence of mental health conditions, with a majority indicating moderate stress (78%) and depression (42%). These outcomes are likely further exacerbated by low access to mental health services in rural regions (Tiberi, 2016; Quran, 2019; Castle, 2020). However, after controlling for factors in multivariable models, the significant indicators of living outside of the home or at work and using makeshift containers to wash hands point to social deprivation factors being the most significant drivers of mental health status, even during a global pandemic. To date, most of the research on the relationship between social determinants and mental health has focused on higher income countries (Maselko, 2017), leaving out the unique contexts of residents and communities like those in this study who face systemic poverty and barriers to accessing adequate resources, including both basic health care needs as well as mental health services.

The pandemic response in rural regions of low- and middle-income nations must consider unique challenges facing the communities, with CBOs being trusted and integral resources to this endeavor. While studies in developed nations have highlighted the efficacy of telehealth services in reducing rural–urban gaps in mental health access and quality of services (American Farm Bureau Federation, 2020; Patel et al., 2020). Low- and middle-income nations such as Haiti may need to implement infrastructural changes first, to allow access to sanitary conditions and basic needs before considering telehealth care or other digital interventions. The outstanding need for broader poverty reduction strategies, including sanitation, clean water access and quality housing in these communities cannot be discounted when considering public health interventions.

This mental health assessment study was the first of its kind to be delivered in the Fontfrede region of Haiti and had meaningful implications for mental health care delivery in the community. In response to the survey findings, capra*care* staff developed and implemented a new mental health program for youths. The study helped identify that the youth were significantly impacted by the COVID-19 pandemic and lost many resources and support systems previously found in schools. Community partnerships like the one used in this study will be essential for COVID-19 response, resiliency and preparedness for future public health emergencies in

Haiti. We recommend that public health professionals and policy-makers identify and prioritize community partnerships with existing, trusted organizations. Just as respondents indicated high levels of awareness of the COVID-19 pandemic, additional health education around prevention and control measures should be communicated through these channels. In rural areas where there are fewer CBOs, the existing and trusted organizations must be partnered with to identify additional effective communication channels and approaches to engage community residents. In some cases, these trusted organizations may be appropriate settings for future health interventions.

Additionally, new research must be funded and developed in low- and middle-income nations in the future, not just during emergencies, and should be embedded in trusted community organizations. Traditional academic research methods may not be appropriate nor effective in reaching the priority population in these communities (Reese and Vera, 2007). Using community members as lay researchers will provide opportunities to residents in this area in terms of economic opportunity and can promote social engagement in areas where isolation may be a challenge. This may be a beneficial way to promote community-level social support as community engagement has been found to strengthen social networks (Shalowitz et al., 2009). Based on the demonstrated effectiveness of health education and health promotion when delivered by peers and members of one's own community, this approach should be prioritized and funded in rural communities.

## Conclusion

This study is the first to provide insights into mental health outcomes after the onset of the COVID-19 pandemic for community members in Les Cayes, a rural region in Haiti. Community members reported lower rates of depression and anxiety when simultaneously having social support. This study may inform public health programs and policies designed to address mental health care and community-embedded health education in rural Haiti and may be further extended to other rural low- and middle-income nations.

**Open peer review.** To view the open peer review materials for this article, please visit http://doi.org/10.1017/gmh.2024.10.

**Acknowledgments.** The authors would like to thank the Capracare, Inc. staff and clients for their contributions to this work. The program described was supported by New York University Behavioral Sleep Medicine (BSM) PRIDE Grant # R25HL105444.

**Author contribution.** Y.G. was responsible for conceiving the study and design, acquisition and interpretation of data, drafting of manuscript, critical revision and final approval of the manuscript. T.P. contributed to the data analysis, interpretation of data and final approval of the manuscript. J.P.-L., T.J., P.B., T.T.I., M.A.P., C.C. and J.R. contributed to critical revision, interpretation of data and final approval of the manuscript. All authors are accountable for all aspects of the work.

**Financial support.** This research received no specific grant from any funding agency, commercial or not-for-profit sectors.

**Competing interest.** The authors report there are no competing interests to declare.

**Ethics statement.** The study was approved by the St. John's University Institutional Review Board (FWA # 00009066).

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
