## [Reviewer Report]

March 24th, 2023

Cambridge Prisms: Global Mental Health

Subject: Submission of research article: 

COVID-19 Knowledge and Mental Health Impact Assessment in Haiti

Dear Prof. Belkin and Editorial Board:

Please find enclosed the research article, “COVID-19 Knowledge and Mental Health Impact Assessment in Haiti,” for consideration with Cambridge Prisms: Global Mental Health. Our manuscript is first to examine COVID-19 knowledge and mental health outcomes in rural communities in Haiti following the COVID-19 pandemic. These communities have long been under-studied, despite being at high risk for many of the world’s epidemics due to the intersection of high-poverty and rapid infectious disease spread. To our knowledge, our study is the first, and to-date only, to examine the mental health and social support impacts of COVID-19 in rural communities in Haiti.

Herein, we describe and report on a study of over 500 participants at a community-based organization in a rural community in Haiti; communicable disease and mental health issues are unique in these communities due to confluences of sanitation, environmental, and political unrest factors faced by residents. We used validated instruments, including the Patient Health Questionnaire (PHQ-9) and Perceived Stress Scale (PSS-4), and included tailored supplemental questions specific to COVID-19 knowledge. Findings revealed notable mental health and social support concerns reported by participants, in this first of its kind study, underpinning the need for investment in community mental health and social programs in rural communities in developing nations. As a direct result of this study, the participating community-based organization implemented a youth mental health counseling program, in response to the identified loss of social connection due to school closures.

Our manuscript aligns with the mission and scope of Cambridge Prisms: Global Mental Health by reporting on the application of mental health assessment in community and real-world settings in a low-income country. These findings will be immediately relevant to the journal’s international readership.

We confirm that this manuscript and results have not been published elsewhere nor are they under consideration with another journal. All authors have approved the manuscript’s content and its submission to Cambridge Prisms: Global Mental Health.

Thank you for your time and consideration of our manuscript. Please do not hesitate to contact me with any questions regarding the research article. 

Sincerely,

Yolene Gousse, DrPH, MPH

Associate Professor 

St. John’s University | College of Pharmacy & Health Sciences

175-05 Horace Harding Expressway 

Fresh Meadows | NY 11365

Email: goussey@stjohns.edu

---

## [Reviewer Report]

The psychological impact of the COVID 19 pandemic in resource-constrained low- and middle-income countries has received less research attention than in high income countries. This paper reports an original investigation in Haiti a severely under-resourced country to establish prevalence of symptoms of depression, anxiety, and other common mental health problems. The authors argue appropriately that their evidence is of relevance to policy makers and program providers in this setting. This study does make an original contribution to knowledge, but in my opinion, there are some significant flaws i in this account of what was done and what was learnt that require revision.

• Careful consideration needs to be given to language use. In particular, it is no longer accepted practice to refer to ‘developing’ countries or nations. This term fails to recognise the deep and sophisticated history that many low- and middle-income countries have. Using the World Bank or other country classification is preferred.

• Line 5 – it was not following the pandemic. In March to May 2020 the pandemic was active and in its early stages and the protections of vaccines were not yet available. The whole paper needs to be read carefully to ensure that it is not conveying the notion that this was a recent assessment. It needs to be acknowledged that what was observed might have been adjustment phenomena that were widespread at the beginning of the pandemic.

• Line 11 – as the assessment did not use a diagnostic measure, the authors cannot say that participants were experiencing depression. I suggest this is replaced with – were experiencing ? some or moderate or clinically significant depressive symptoms.

• Lines 49 – 53. The prevalence data are presented completely uncritically. There is no engagement with how they were generated and the potential impact of methodological differences. In one place they say that prevalence of depression in Haiti has doubled to 5.82% and a line later they say that 50% were depressed. There needs to be much more comprehensive engagement with these disparities and what they mean. Detailed discussion is needed of how the data were generated and whether any of the of the psychometric properties of the instruments had been formally validated against a gold standard. If not this should be added as a significant study limitation.

• Line 62 refers to disparities in health in Haiti, but the reader is not told what these are, including in mental health and what they are attributed to. This needs to be added.

• Line 62 Raw numbers are not helpful to people who do not know Haiti – population fractions are more comprehensible. Please convert these to percentages or fractions.

• Line 69 reports, without describing the evidence that the least advantaged people had the highest rates of death by suicide during the COVID pandemic. The authors must provide the evidence to support this assertion and where and how the data were derived.

• Line 78 – 80 – the authors need to explain what the comparator is otherwise they cannot claim that it is linked to COVID-19.

• Line 98 – makes causal assertions about infections being attributable to lack of access to care. This is probably not a causal relationship – the infections occur, but it is lack of access to treatment that contributes to the burden ? of untreated infections and communicable diseases. More detailed engagement with causal pathways, including of the impact of poor quality and crowded housing, is needed.

• Line 111 – The Methods need to be described with sufficient precision and detail to enable replication. This would not be possible from the account provided here. Please describe in detail what ‘canvas[ing]’ in Fonfrede and Les Cayes to identify potential participants actually involved and how participants were identified and invited to join the study.

• Line 121 – it is not clear what ‘Instrumentation’ means. Please describe how the survey instrument was developed, whether the questions were standardised or study-specific. How was comprehensibility established and culturally verified?

• Line 155 Please provide a reference for the ‘PHQ-4 subscales’ and for any evidence of local validation. It appears that the PHQ was altered to make it COVID specific. Please describe this process.

• Line 167 Please spell out the name of the PSS in full and provide a reference for it. Please describe how it was adapted to be COVID specific

• There is no ‘Procedure’ section and therefore no description of how the data were collected and, if not self-reported, by whom. This needs to be added.

• The analysis is at a basic level, please seek the advice of a biostatistician and undertake a multivariable analysis

• Line 255 ‘never unconfident’ the double negative makes it difficult to know what is being said. Please rephrase so that there is one dimension.

• Cannot attribute the direction of an association in a cross-sectional survey. The authors need to offer both directions of this associations – that the psychological distress was a consequence of COVID or the alternative that more psychologically distressed people experienced the pandemic in a different way to those who were not distressed.

• Although they cite Lund’s evidence about the social determinants of mental health problems, they do not question the use of the term ‘mental illness’ to describe psychological distress arising in the context of severe adverse circumstances or acknowledge that it could be construed as a normal response to adversity

---

## [Reviewer Report]

RE: Manuscript Review - GMH-23-0071 “COVID-19 Knowledge and Mental Health Impact Assessment in Haiti”

Dear Review Committee Members,

I hope this email finds you well. I am writing to express my sincere gratitude for the time and effort the journal review committee has dedicated to reviewing our manuscript titled “COVID-19 Knowledge and Mental Health Impact Assessment in Haiti,” which we submitted to Cambridge Prisms: Global Mental Health.

We truly appreciate the insightful comments and constructive feedback provided by the esteemed reviewers. Your thoughtful observations have proven invaluable in improving the quality and clarity of our research. As requested, we have thoroughly reviewed all the comments and have addressed each one meticulously in the revised manuscript.

We believe that the revised version of our manuscript now reflects a stronger and more impactful contribution to the knowledge base surrounding the intersection of COVID-19 and mental health in Haiti.

Also, we are submitting the response to reviewers' comments, and two (in tracked changes and clean) versions of the revised manuscript incorporating all the changes and updates based on your valuable feedback. 

Thank you for your continued support, and we look forward to hearing from you soon.

Best regards,

Yolene Gousse, DrPH, MPH

---

## [Reviewer Report]

The revision of this paper has addressed some of the concerns raised in the first round of reviews, but some have not been addressed and other problems have been introduced.

The authors write with sensitivity about the impact of poverty on mental health and describe the extremely difficult circumstances of peoples’ lives in Haiti well. The impacts of the pandemic on household livelihoods and incomes are evoked vividly.

I have several concerns, most about the methods of the study and the unqualified confidence with which the findings are presented. There are aspects of the methods that could not be replicated from the account provided here:

• Haiti is a lower-middle income country by the World Bank classification. This needs to be corrected from ‘low to middle’ throughout and the appropriate reference to the World Bank Group country classification added.

• The Introduction needs to end in a clearly stated aim or aims and not with a statement of what was done, this needs to be corrected.

• Line 149 It is not clear what ‘domestic composition and structural factors’ mean? – please provide a definition

• Please describe the inclusion criteria and recruitment strategies specifically in the Methods and, on the basis of these, consider potential selection biases and their implications in the Discussion.

• What does ‘recorded completely’ mean – were the interviews audio recorded, or documented in field notes? Please describe this process precisely?

• The authors say that no changes were made to the PHQ-4 , but the phrasing of the questions were altered to link them to COVID – this is a non-standardised change and it cannot be assumed that the psychometric properties are the same or directly comparable to its standard use, this limitation and its potential impact on the findings needs to be acknowledged and considered in detail in the Discussion.

• No effort is made to compare the characteristics of the study sample to the general population of Haiti. Accepting that this might be difficult, it is crucial for consideration of generalisability. Please add a table making these comparisons or explain that it is not possible.

• Line 289 never or almost …? a word missing here

• Concludes legitimately that there is a need for more services, but does not elaborate on what these might be ? provision of food, housing, access to sanitation and income-generating work and not only to mental health care. Please provide a more elaborated account here

• Makes claims without supporting evidence that the interventions provided by the CBO are effective. Please provide the evidence to support these claims.

---

## [Reviewer Report]

Dear Dr. Belkin and Editorial Board: 

I am writing to express my sincere gratitude for your favorable decision to publish our article and the time and effort the journal review committee has dedicated to reviewing our manuscript titled “COVID-19 Knowledge and Mental Health Impact Assessment in Haiti”. 

As requested, please find attached two copies of the manuscript. The first version is in tract changes and addresses the reviewers’ revisions; and the second version is a final clean copy to advance the paper for printing and publication. 

Thank you for your continued support.

Best regards,

Yolene Gousse, DrPH, MPH

---

## [Reviewer Report]

Thank you for responding to the suggestions in the reviews. This account of the study is considerably improved.

Minor amendments are still required:

The study is referred to as a ‘pilot’ only at the end of the Discussion. It appears to be substantial - 500 participants and is not referred to as a pilot elsewhere in the paper. Please either remove this descriptor or explain early in the paper that it is a pilot with an elaborated description of what it is a pilot for. Early in the paper (Line 159) we are told that the survey was pilot tested before implementation so the study as a whole does not appear to be accurately described as a pilot.

This research identified social factors as the main risks for the mental health problems that were documented. However, the only implication that is drawn out of the findings is for increases in mental health care and counselling. It is imperative that the need for strategies for poverty reduction, improvement of housing quality, access to clean water and promotion of social connections are considered and discussed because individual counselling is very unlikely to have an impact on these.

---

## [Reviewer Report]

Dear Editorial Team:

We appreciate the close review to our manuscript ‘COVID-19 Knowledge and Mental Health Impact Assessment in Haiti.’ Please find enclosed a revised manuscript and below responses to reviewer comments. We believe the revisions should adequately address the outstanding comments.

Please do not hesitate to be in touch with any additional questions regarding the manuscript.

Sincerely,

Yolene Gousse, DrPH, MPH